# Optimal control prevents itself from eradicating stochastic disease epidemics

**Rachel Russell** (ID)\*, **Nik J. Cunniffe** (ID)

Theoretical and Computational Epidemiology, Department of Plant Sciences, University of Cambridge, Cambridge, United Kingdom

\* rjt42@cam.ac.uk

## Abstract

The resources available for managing disease epidemics – whether in animals, plants or humans – are limited by a range of practical and financial constraints. Optimal control has been widely explored for optimising allocation of these resources to maximise their impact. The most common approach assumes a deterministic, continuous model to approximate the epidemic dynamics. However, real systems are stochastic and so a range of outcomes are possible for any given epidemic situation. The deterministic models are also known to be poor approximations in cases where the number of infected hosts is low – either globally or within a subset of the population – and these cases are highly relevant in the context of control. Hence, this work explores the effectiveness of disease management strategies derived using optimal control theory when applied to a more realistic, stochastic form of disease model. We demonstrate that the deterministic optimal control solutions are not optimal in cases where the disease is eradicated or close to eradication. The range of potential outcomes in the stochastic models means that optimising the deterministic case will not reliably eradicate disease – the required rate of control is higher than the deterministic optimal control would predict. Using Model Predictive Control, in which the optimisation is performed repeatedly as the system evolves to correct for deviations from the optimal control predictions, improves performance but the level of control calculated at each repeated optimisation is still insufficient. To demonstrate this, we present several simple heuristics to allocate control resources across different locations which can outperform the strategies calculated by MPC when the control budget is sufficient for eradication. Our illustration uses examples based on simulation of the spatial spread of plant disease but similar issues would be expected in any deterministic model where infection is driven close to zero.

## Author summary

Improving epidemic outcomes in situations with limited resources for control is important across human, animal and plant health. An example of this type of optimisation

**Data availability statement:** Implementations of the model and control algorithms are

available at https://github.com/racheltrimble/plant_disease_model_oc_paper (commit hash: b52801e).

**Funding:** RR and NJC acknowledge the Engineering and Physical Sciences Research Council and Defra for support via the AgriFoRwArdS CDT [EP/S023917/1]. The funders had no role in study design, data collection and analysis, decision to publish, or preparation of the manuscript. For the purpose of open access, the author has applied a Creative Commons Attribution (CC BY) licence to any Author Accepted Manuscript version arising from this submission.

**Competing interests:** The authors have declared that no competing interests exist.

is deciding which infected plants to remove first to reduce ecosystem or agricultural losses from an invasive plant disease when the budget for control is limited. A common approach to these optimisations is to model the epidemic as a totally predictable (deterministic) process and modern algorithms allow these optimisations to scale to large, real-world problems (e.g. allocating resources across a landscape). However, when it is possible, the agencies responsible for allocating resources generally prefer solutions where the disease is removed entirely from the system or from a subset of the population. We tested solutions from deterministic optimisations against an epidemic model which includes some of the random effects we would expect to see in real applications. We showed that, in cases where eradication is feasible, the optimisations which use deterministic models are not optimal. They do not account well for the range of outcomes that can occur and so they can be outperformed by much simpler "rules of thumb". This helps managers and researchers understand where the deterministic models are appropriate and where optimisations need to use different methods to account for the randomness of real epidemics.

## Introduction

Epidemic modelling is well established across human, animal and plant health as a way to predict how disease will spread within and between populations [1–4]. The most common approach is compartmental modelling where the population is divided into mutually exclusive groups depending on disease status and other factors affecting transmission or recovery (e.g. age or location). Transitions between the compartments are most often modelled using either deterministic, continuous ordinary differential equations (ODEs) or as a stochastic process where discrete individuals move from one compartment to another according to a Poisson process. Deterministic, continuous models are more analytically tractable but stochastic, discrete models are more able to represent disease features such as pathogen eradication [5].

The targeted outcome for actions carried out to control an epidemic vary dependent on the scenario and pathosystem in question (see [6] for a review of the equivalent problem in the context of established invasive species). For plant diseases, protecting key areas of agricultural productivity or biodiversity may be all that is feasible in some circumstances [7]. However, if control is started sufficiently quickly and eradication is possible – either across an entire region of interest or within some more limited geographical areas – it is the preferred approach due to vastly lower mortality and costs [8]. Examples in the UK can be seen in responses to an invasion of Asian Longhorn Beetle [9], in the plans for Tomato brown rugose fruit virus [10] and for *Xylella fastidiosa* [11]. Eradication programmes can be sustained over some time, for example the failed eradication of citrus canker from Florida from 1998-2006 [12] and the attempt to eradicate Xylella fastidiosa on almond from Alicante in Spain which has been ongoing since 2017 [13]. However, there are management scenarios where eradication is not feasible due to financial and social costs, for example in the case of oak processionary moth in the south of England [14] the pest had become established across a large area of busy public parks in London before it was detected. Eradication may also be prevented by ongoing infection from sources outside the managers' control, for example in the control of wheat rust control in Ethiopia [15]. A wide range of studies have been carried out which evaluate control measures by using epidemic models. The control measures range from simple heuristics to complex machine learning based methods (heuristics: [16,17], optimised heuristics: [3], genetic algorithms: [18], simulated annealing: [19], MDP based: [20], machine learning: [21]).

Optimal control is a family of techniques for calculating the best set of time varying controls to apply to a dynamic system. The "best" control is defined by a reward function to be maximised which usually takes the form of an integral of some feature of the system over time. For example, in this work, we defined a reward function as the integral of the number of healthy hosts in a population. The field can be broadly divided into stochastic optimal control where the system is modelled as a Markov Decision Process (MDP) and deterministic, continuous optimal control where the dynamics are modelled as a series of linked differential equations. In an MDP, the system is modelled as a collection of states. For example, the states in a simple SIR model might be the numbers of infected and susceptible hosts in the population at that time. The system moves between states stochastically with the transition probabilities dependent on the underlying dynamics and the controls applied to the system [22,23]. Given the stochastic nature of the epidemic control problem, it would seem appropriate to use MDP based approaches. However, solving these problems exactly scales poorly with system size and is computationally intractable for realistic epidemic models (discussed further in Context). A separate approach to the stochastic optimisation is to find a useful approximation using techniques such as genetic algorithms [24] or reinforcement learning [21]. However, these methods provide no guarantees around the optimality of their solutions and so are not directly comparable. Deterministic, continuous optimal control problems can be solved using a range of approaches usually classified as direct methods, indirect methods and dynamic programming. However, for many real life problems, the analysis for any of these formulation is intractable and they are more commonly solved by quantising the system in time and solving using a Non-Linear Programming (NLP) solver [25].

Model Predictive Control (MPC) was developed to extend the application of various deterministic optimisation techniques (including optimal control) to more realistic scenarios – such as epidemic modelling – where the deterministic model being used is not a perfect representation of the real system or where there is noise or uncertainty in the measurement of the system state. When using MPC, the optimisation is run repeatedly as the system evolves, using the latest status of the system to set the initial conditions for each iteration of optimal control. Optimal control has a single horizon defining the future time period the calculation will optimise over (the optimisation horizon). MPC has a second horizon determining how often the optimisation is rerun (the control horizon). Rerunning the optimisation allows the controller to respond to unpredicted deviations [26].

Continuous, deterministic optimal control has been widely proposed as a solution to optimisation of epidemic control across different applications, host taxa, underlying compartmental models and proposed cost functions. The simplest applications focused on small compartmental models (susceptible and infected hosts only) with a trade off between the costs of infectious hosts and the costs of treatment. In these cases, the optimal approach is to use bang-bang control (where each control is either at its maximum or minimum value) to reach an endemic level of infection where costs of disease optimally balance costs of control as soon as possible (e.g. [27]). This was extended to models with more compartments [28], metapopulations [29,30], different forms of control (e.g. level of mixing restrictions in [31] vs treatment and inoculation in [27]) and different cost functions [32,33]. Many papers present results where eradication or infected numbers close to zero is the goal or result of the presented strategies without consideration of the stochastic nature of the problem for both single populations [34–37] and metapopulations [30,38]. [39] looks at a simple metapopulation with two subpopulations but scenarios where eradication is feasible or close to feasible are explicitly excluded as being too simple to be interesting.

In summary, optimising control of epidemics by modelling them as deterministic systems and applying optimal control theory has often been used to understand how control of epidemics can be improved. However, these methods require the use of numerical methods to scale to more complex, real life scenarios and the applications of these methods to date have not addressed inaccuracies that could occur due to stochastic eradication. Hence, this paper seeks to address the following questions:

1. How should we expect stochastic population and metapopulation systems controlled with control trajectories derived from deterministic models to behave in cases where eradication is possible?
2. How does the performance of these optimal controls compare to simpler priority based heuristics which prioritise eradication?
3. How do these results change as we start to scale the number of subpopulations in the metapopulation system?

## Materials and methods

In this section, we describe the case study we have used, our approach to implementing the optimal control and how this was combined with model predictive control. We then describe the priority based control approach used as a reference and finally give details of the individual model parameterisations used in the different comparisons. For further details, the code used to generate all results can be found at: https://github.com/racheltrimble/plant_disease_model_oc_paper

### Problem setting

In order to explore the performance of optimal control in both eradication and ongoing management contexts, we created a simple metapopulation model which captures the main behaviours of a plant disease epidemic spreading within and between spatially separated subpopulations. In the context of plant disease, infected trees are removed by culling (also referred to as "roguing", [40]) to prevent disease spread to other neighbouring hosts. Healthy hosts can also be thinned to slow spread [41]. Hence, the control actions for this system are the rate of culling of infected hosts and the rate of thinning of susceptible (healthy) hosts per subpopulation at each control step. The aim of the control is to maximise the "yield" of the system i.e. the integral of susceptible hosts with respect to time. The number of management actions in a given control step was constrained by a fixed budget per control step with a fixed culling and thinning cost per host removed.

We evaluated optimisation approaches by running them on a stochastic, spatiotemporal metapopulation simulation with transitions between compartments modelled as Poisson processes. The model used a standard SIR compartmental model [5] for the dynamics within each of the subpopulations. We assumed that disease progress is fast compared to host lifetimes and so host births and non-pathogen-related deaths were ignored. We simulated the model using the Gillespie algorithm [42]. The differential equations for each of the subpopulations for the equivalent deterministic system were:

$$\frac{dS_x}{dt} = -\Gamma_x S_x - T_x, \qquad \frac{dI_x}{dt} = \Gamma S_x - \gamma_x I_x - C_x, \qquad \frac{dR_x}{dt} = \gamma I_x + T_x + C_x,$$

where $S_x$, $I_x$ and $R_x$ are the number of hosts in each compartment in subpopulation $x$ (i.e. assuming density dependent transmission), $C_x$ is the net rate of culling in subpopulation $x$,

$T_x$ is the net rate of thinning and $\Gamma_x$ is the force of infection on that subpopulation. $\Gamma_x$ was calculated as:

$$\Gamma_x = \beta_{int}I_x + \beta_{ext}\sum_{y=1,y\neq x}^{P} k(d_{xy})I_y,$$

where $P$ is the number of subpopulations in the system, $\beta_{int}$ and $\beta_{ext}$ are the infection rates within and between subpopulations and $k(d_{ab})$ is *a* kernel function representing how infectivity scales with distance between subpopulations $a$ and $b$. We modelled aerial dispersion based on the pairwise distance ($d$) between each of the subpopulations using *a* power law spatial kernel ($k(d_{ab}) = d_{ab}^{-3}$). We selected the kernel arbitrarily but we note it is in line with dispersal kernels as parameterised for various plant pathogens [43]. Note that both the underlying epidemic and the controls are modelled as random processes in continuous time but the control decision intervals are discrete. This aligns to the quantised version of the optimal control problem and makes the plausible assumption that human decision making happens at discrete intervals. The use of a random process with a controllable rate for host removal (as opposed to removing hosts at the start of each discrete control window) means there is a delay between the controller requesting that hosts are removed from the system and the host being removed. This mimics delays between decision making and control in the real system. We set the control step for the environment to 1 time unit (t). The detailed parameterisations and scenarios used for testing are described in Simulation details.

## Optimal control

All optimal control simulations were based on numerical solutions to the direct formulation of the relevant problem. In the direct formulation of optimal control, the dynamic system is discretised in time and the reward function is optimised directly using an NLP solver. We ran two variations of the optimal control calculations – proportional and absolute – with the intent of exploring different ways to allow the optimised control to approach closer to zero infected hosts. We used a constant cost per host for control actions. The optimal control used an integral reward rather than reward based on final healthy host population because we expected this to be a more pragmatic measure in the context of a less predictable, stochastic model. Additionally, we set the bounds on the control per subpopulation with limited spend per subpopulation. This is not unrealistic and, pragmatically, allowed for faster convergence of the optimisation. The NLP problem was solved using ipopt [44] using the HSL MA97 linear solver. The optimal control code was implemented in Python using the Casadi library [45].

**Proportional variant.** The system equations and constraints for the proportional variant were:

$$\frac{dS_x}{dt} = -\Gamma_x S_x - T_x S_x, \qquad \frac{dI_x}{dt} = \Gamma S_x - \gamma I_x - C_x I_x,$$

$$B = \sum_{x=1}^{P} C_x I_x m + T_x S_x n,$$

$$S_x \geq 0,$$

$$I_x \geq 0,$$

$$0 \leq C_x \leq 1,$$

$$0 \leq T_x \leq 1,$$

$$0 \leq B \leq 1,$$

where $m$ and $n$ are scaling factors for culling and thinning costs respectively. Note that the constraints on $C_x$ and $T_x$ are a limit on the maximum rate of control actions per subpopulation in each time period.

The objective function expressed as a reward to be maximised was defined as:

$$J = \int_{t=0}^{T_{OC}} \sum_{x=0}^{P} S_x(t),$$

where $T_{OC}$ is the optimisation horizon (determining how far in the future the reward is integrated over). We did not use discounting in the reward function because we are using a fixed horizon and so there is no requirement for another mechanism to devalue rewards in the future. This overall formulation is called **proportional control** in the remainder of the text. In the simulated environment, the cumulative reward was incremented by $S$ at the start of each control interval and so can be summed to give a sampled version of the integral reward for the optimal control. We set $T_{OC}$ to allow *a* large proportion of the uncontrolled epidemic trajectories to run to completion and aligned this to the simulation duration ($T_{sim}$).

We implemented the stochastic environment with the control specified in terms of a rate of controlling hosts during that control step. This is not directly compatible with the controls generated by the proportional control which are in terms of a proportion of the current population. If the proportional control trajectories were integrated with the stochastic model by multiplying $C_x$ and $T_x$ by the current per-compartment populations, the populations would sometimes be higher than the deterministic model had predicted and the budget constraints could be violated. Hence, the rates were multiplied by the number of hosts predicted by the optimal control model at the start of each control period.

**Absolute variant.** The absolute variant of the optimal control used the same underlying epidemiological model but did not assume that the rate of control was proportional to the current population. For **absolute control**, the equations were:

$$\frac{dS_x}{dt} = -\Gamma_x S_x - U_x, \qquad \frac{dI_x}{dt} = \Gamma_x S_x - \gamma I_x - D_x,$$

$$B = \sum_{x=1}^{P} D_x m + U_x n,$$

$$S_x \geq 0,$$

$$I_x \geq 0,$$

$$0 \leq D_x \leq 1,$$

$$0 \leq U_x \leq 1,$$

$$0 \leq B \leq 1.$$

Hence, in both formulations, the budget constraint is in terms of the number of hosts (i.e. $C_x I_x = D_x$) and the units of $m$ and $n$ are the same in both formulations (cost time per host). However, the within control step dynamics are different.

The control step duration was set to 1 to align to the control interval in the stochastic environment. The ODEs were solved at a resolution 6 times faster than the control resolution in order to allow stable convergence. The IPOPT solver requires a starting point for the optimised variables in order to initialise the optimisation. The control was initialised to zero and the quantised state variables intialised to be consistent with an integrated solution to the state equations (using an RK4 integrator).

## Model Predictive Control (MPC)

For the MPC work, we set the optimisation horizon to $T_{OC}$ (consistent with the main optimal control tests). Initially, we set the controller to rerun the optimisation every 4 control steps (control horizon) as a balance between control response and runtime. We also present a variant with a control horizon of 1 to enable a fairer comparison to the priority based control where both controllers receive information about the system at the start of every control step.

## Priority based controls

We ran several simple, alternative controls to compare against the optimal control solutions. Each of the alternative controls creates a priority list to determine which subpopulation is allocated treatment resource. The alternative controllers only ever perform culling as thinning is never directly beneficial from the point of view of eradication. The controller removes all the infected hosts from the highest priority subpopulation, then moves successively on to the next highest priority subpopulation and repeats the allocation until the budget is used up or all infected hosts have been culled. The different variants tested were:

- **Prioritise I**: Prioritise based on number of I hosts in the subpopulation.
- **Prioritise S**: Prioritise based on number of S hosts in the subpopulation.
- **Random subpop**: Prioritise randomly with the prioritisation order rerandomised at each control step.
- **Random subpop one shot**: Prioritise randomly but only chose once and keep the prioritisation order constant.
- **Equal per host control**: An unprioritised baseline where resources are allocated evenly between culling and thinning and between the different nodes proportional to their initial populations.

This approach assumes immediate culling when deciding how much resource to assign to each subpopulation and does not take into account any increase or decrease in infected hosts during the control step. However, any remaining budget after the controller reaches the end of the priority list is split across the subpopulations proportional to the amount of culling resource already allocated. These controllers are simple and fast and do not require knowledge of the system dynamics. In terms of drawbacks, they require regular updates on the system status. However, in reality, hosts would need to be surveyed before culling activities in order to work out where the diseased hosts were located.

## Simulation details

**Deterministic controls applied to a simple metapopulation model.**   We initially consider a square system of 2 by 2 subpopulations with the initial population sizes per node randomly selected from the range 0 to 200 (uniform distribution) but fixed for each scenario (Fig 1). We started each simulation with a small number of infected hosts in one subpopulation such that eradication was still feasible with a moderate level of control resource. We parameterised the control for the simulated epidemic so it could be well controlled by focusing on removing the infected hosts as soon as possible (called the **controllable system** in the rest of the text) and, separately, making the culling and thinning actions 3 times as expensive (the **challenging system**, see also Table 1). Our default parameterisations gave an epidemic that would usually kill a large fraction of the hosts in the absence of control (Fig 1b). We set the culling and thinning costs so that the controller could cull or thin up to 1/8th of the population per control step in the controllable case and 1/24th of the population in the challenging

case. We set the infection rates ($\beta_{int}$ and $\beta_{ext}$) so the force of infection from an infected host in a neighbouring node would be around 20% of one in the same node for the typical metapopulation spacing used (1 unit). The topography of the sub-populations in the 2x2 subpopulations and the corresponding dynamics in the absence of control are shown in Fig 1. We evaluated the performance of the different controllers by comparing the distribution of 100 simulation runs. Mean performance could be misleading for bimodal distributions common in epidemic simulation outcomes when stochastic extinction is possible and so we have not reported it.

We used the 2x2 controllable system to compare the proportional and absolute control variants. In proportional control, the per subpopulation bound represents the maximum proportional rate of hosts in each subpopulation that can be culled or thinned in each control step. In absolute control, this is expressed as a proportion of the overall population in the largest node. Hence, the subpopulation bounds for the absolute control are less stringent than

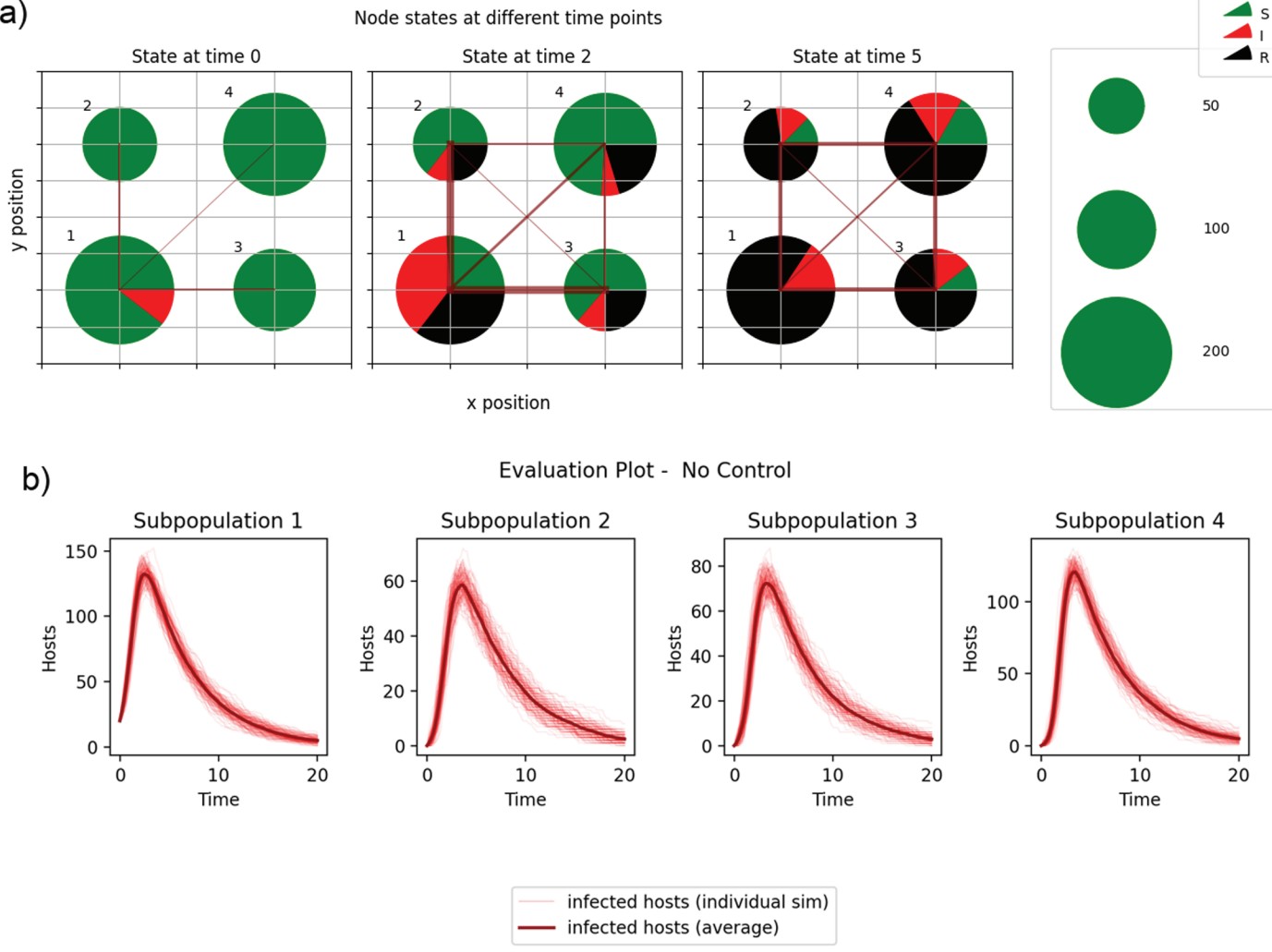

**Fig 1. Layout of simple system used for testing.** a) The thickness of the connecting line indicates the mean force of infection between the two subpopulations. b) Progress of an epidemic without control (infected hosts against time for each subpopulation) simulated 100 times.

**Table 1. Epidemic, simulation and control parameterisations for the 2x2 and 4x4 systems. Co = Controllable, Ch = Challenging, Int = Intermediate. Units are all abstract but included to clarify dimensionality (t = time). *The number of hosts in each subpopulation was randomly generated so expressing the parameterisation of costs as shown in the optimisation description is not meaningful. Translating between the notations:** $n = m = \frac{B}{p \sum_{x=1}^{P} N_x}$.

| Parameter | Symbol | Values | | | | | Units |
|---|---|---|---|---|---|---|---|
| Rate of secondary infection within subpopulations | $\beta_{int}$ | 0.01 | | | | | hosts$^{-1}t^{-1}$ |
| Removal rate of uncontrolled infection | $\gamma$ | 0.2 | | | | | $t^{-1}$ |
| Max hosts per metapopulation | $N_{max}$ | 200 | | | | | hosts |
| Duration of simulation | $T_{sim}$ | 20 | | | | | t |
| Optimal control horizon | $T_{oc}$ | 20 | | | | | t |
| **System sizing** | | **2x2** | | **4x4** | | | |
| Rate of secondary infection between subpopulations | $\beta_{ext}$ | 0.002 | | 0.00105 | | | hosts$^{-1}t^{-1}$ |
| Total initial infected hosts | $I_0$ | 16 | | 64 | | | hosts |
| Initial infected subpopulations | $P_{I0}$ | 1 | | 4 | | | populations |
| **System budget** | | **Co** | **Ch** | **Co** | **Int** | **Ch** | **Units** |
| Proportion of hosts that can be controlled per cycle* | $p$ | $\frac{1}{8}$ | $\frac{1}{24}$ | $\frac{1}{8}$ | $\frac{1}{16}$ | $\frac{1}{24}$ | N/A (ratio) |

for the proportional control and we would expect absolute control to perform at least as well or better than the proportional control.

**Comparison to priority based control.**   We compare MPC with the prioritised control algorithm across both controllable and challenging systems. These experiments focused on MPC-based solutions because comparing optimal control (where control is determined based on a single, initial state measurement) with prioritised control (where the state of the system is assumed to be known at every control interval) is not a fair comparison. We used the absolute optimal control formulation for the MPC based on analysis of results from the initial trials. Hence, we tested the following control variants:

- **MPC absolute horizon 1**: MPC absolute with control horizon of 1.
- **MPC absolute**: MPC absolute with control horizon of 4.
- **Random subpop**: Priority based control using randomly selected priorities at each control step.
- **Random subpop one shot**: Priority based control using priorities selected once at the start of simulation.
- **Prioritise I**: Priority based control using the absolute number of infected nodes in a host as the prioritisation criteria.
- **Prioritise S**: Priority based control using the absolute number of susceptible nodes in a host as the prioritisation criteria.
- **Equal per host control**: an unoptimised baseline.

**Scaling to larger metapopulations.** Finally we applied the optimal control and priority based control to a larger 4x4 grid system to demonstrate that the issues described are not limited to trivial examples. We scaled the 4x4 systems to be approximately equivalent in "difficulty" to the 2x2 systems by scaling the costs of culling and thinning to allow the same level of control per host. We increased the number of initially infected hosts proportional to the number of subpopulations of the system (to allow a larger budget for control without the initial infected population being eradicated trivially). The initial infected population was spread over a subset of the subpopulations in the larger system in order to maintain a mixture of within and between subpopulation spread at the start of each scenario. Finally, we scaled the external rate of infection inversely proportional to the average level of connectivity in the system ($c$) such that $\beta_{ext}c$ was consistent across the different systems where:

$$c = \frac{1}{P}\sum_{x=1}^{P} \sum_{\substack{y=1 \\ x \neq y}}^{P} k(d_{xy}).$$

This means that the overall levels of transmission across the system remain similar in spite of the increased number of possible sources of infection. We also added an intermediate parameterisation of control where there was sufficient budget to apply control to 1/16th of the population in each control step to look at behaviours where eradication was reachable but more difficult to attain.

## Results

### Deterministic controls applied to a simple metapopulation model

We start by analysing the behaviour of deterministic optimal control (without MPC) when it is applied to a small stochastic system with low control costs (2x2 controllable system, Fig 2). With low control costs, the controller can afford a high rate of culling and we would expect the disease to be eradicated in the first few cycles of the simulation. However, both variants of optimal control allow the disease to persist beyond the end of the simulation for a large fraction of the simulations ($N << 1000$ in the eradication time plot) and hence these controllers achieve low rewards. Additionally, we see that the performance of the proportional control is better than that of the absolute control. This is unexpected because the per subpopulation bounds on culling and thinning are more restrictive in the proportional case. Any allowable strategy for the proportional control is also allowed within the wider bounds of the absolute control and so the absolute variant would be expected to perform better.

These results can be explained by comparing the state trajectories generated by the deterministic model internal to the optimal control with the corresponding, stochastic, simulated trajectories (Fig 3). The control with the wider limits (absolute control) calculates that it is possible to cull sufficient infected hosts to reduce the levels of infection to zero. However, when this control is applied to the stochastic system, the average infection levels are driven close to zero but, in a significant fraction of cases, the disease is not eradicated and is able to reestablish. The culling rate selected by the optimal control is optimised based on the deterministic model to remove all the infected hosts in the first cycle but it is constrained so it cannot generate controls that lead to negative numbers of infected hosts. Hence, the optimiser will pick a rate that leads to exactly zero infected hosts in the deterministic model at the end of the control step. The optimal control will not select a larger rate to ensure eradication even when budget is available because, in the deterministic model, it is unnecessary and larger rates

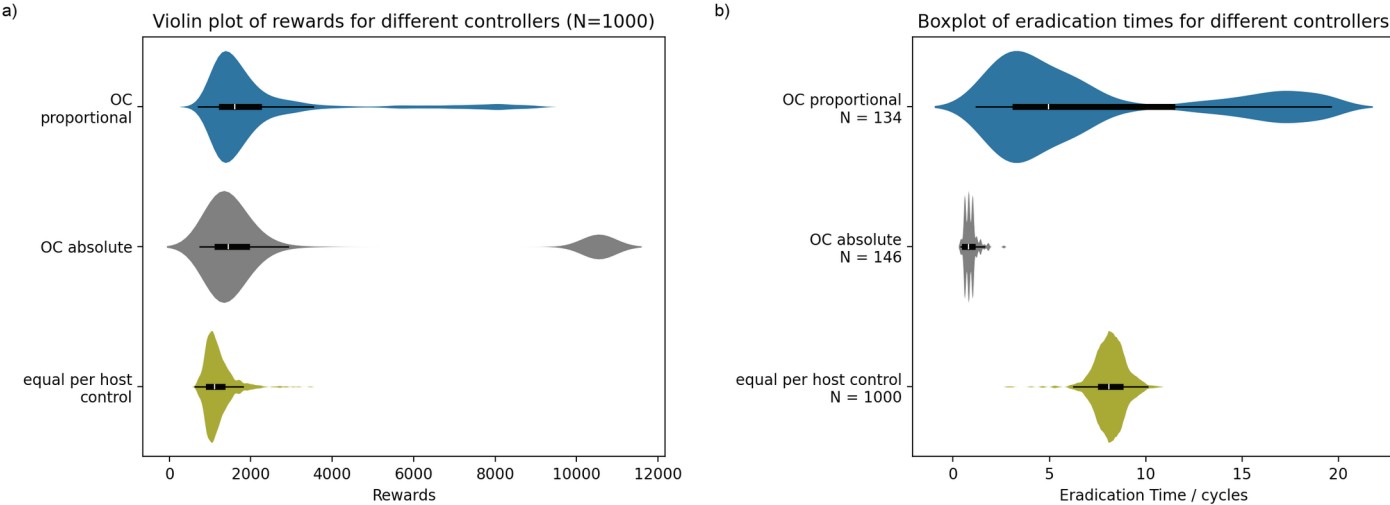

**Fig 2. OC controller performance for the controllable 2x2 system.** Violin plots of a) rewards and b) eradication times. Counterintuitively, the more constrained proportional control outperforms the less constrained absolute control. The height of the plots is scaled separately for each of the controllers. The central black box in each violin shows the data quartiles and the whiskers show the farthest datapoint within $1.5 \times IQR$ of the upper and lower quartiles. 1000 simulations were run for all experiments on the eradication time plot but the number of samples included in the plot ($N$) is shown separately for each of the controls because the disease was not eradicated in all simulations.

would result in $S < 0$ which violates the supplied constraints. If this rate is applied to an equivalent stochastic system, disease is only eradicated in a proportion of the simulations and the average number of infected hosts remains above zero.

In comparison, the proportional control is prevented from spending sufficient budget on each subpopulation to ensure local eradication and it adopts a more conservative strategy including thinning to make the system more robust to disease spread later in the simulation. In the deterministic model, it performs worse than the absolute control as we would expect. However, it is marginally more successful than the absolute control when applied to the stochastic system because, for the selected control, the levels of infection are high and so the deterministic model is a better approximation to the stochastic system. Hence, the more heavily constrained proportional control gives a better outcome but it cannot be described as optimal in any meaningful sense.

## Comparison to priority based control

Figs 4 and 5 summarise the results for the 2x2 controllable and challenging systems comparing MPC and priority based controls. We used the absolute variant of MPC for this comparison (MPC absolute and MPC absolute horizon 1) as it is the least constrained. Priority based controls outperform the MPC controllers in the controllable case but both OC and MPC are beneficial in the challenging case. This is consistent with the results from the initial testing; when eradication is feasible, repeating the optimal control calculation does not fix the underlying issue because each iteration of optimisation systematically underestimates the level of control required. Decreasing the interval between recalculating the optimal controls (MPC horizon absolute horizon 1) improves performance as expected for the controllable system but shows minimal benefit for the challenging system indicating that the deterministic model is a good predictor of the disease trajectory when eradication is not feasible.

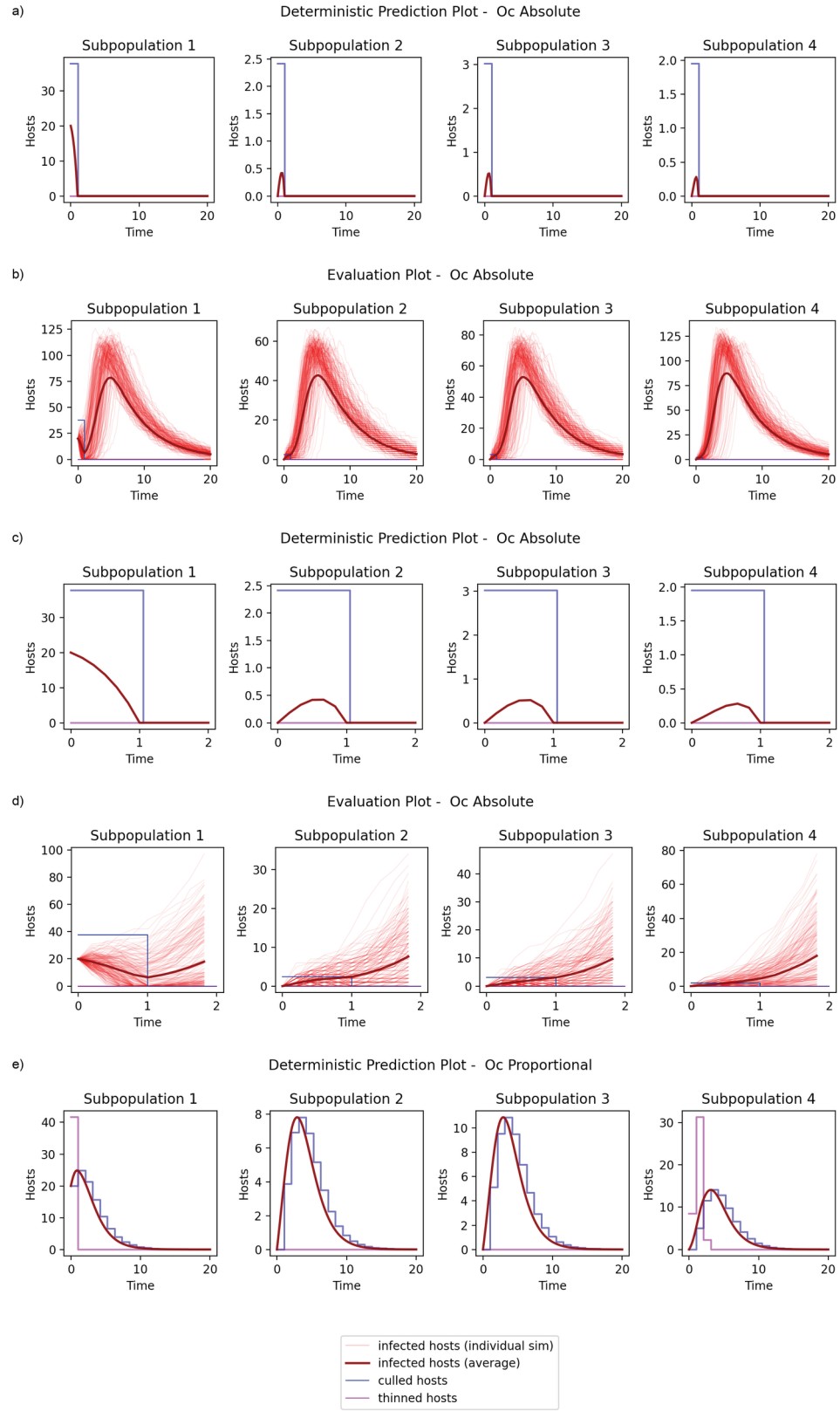

**Fig 3. Comparing the state predictions of the absolute optimal control variants for the controllable system with the results in the equivalent stochastic system.** a) State predictions for the controllable system. b) Results from the stochastic system. (c) and (d) show the same comparison but focusing on the first two control steps. For the absolute

case, the optimal control predicts that the disease will be eradicated and no further effort is required but the stochastic plots demonstrate that this is not reliable in the more realistic model and the disease persists to the end of the simulation in a large number of trajectories. Looking at the state predictions of the proportional optimal control variant (e), it is unable to expend enough resource to eradicate the disease in the first control step and so adopts a more conservative strategy with a portion of the initial resource expended to thin the population (shown in purple) to reduce the longer term impact of the epidemic. Note that the y-axis scales are different for the 5 subplots.

The superior performance of optimal control based methods on the challenging system does not detract from the main point of our argument because, away from the eradication scenario, the approximations in the optimal control are valid and we are not making any claims that the priority based methods are optimal for all systems. However, the results for the challenging system are not hugely different between the approaches. The overall benefit of any of the optimised methods is relatively small compared to the equal per host baseline and the priority based methods still eradicate the disease earlier. For the priority based control, there is no obvious advantage to one method of ranking over another indicating that the main benefit of the priority based methods is from the full allocation of resources rather than effective evaluation of the risk posed by different subpopulations. We can see from the example simulation traces for the controllable system that the MPC culling control tracks the number of infected hosts with a small positive offset. This is consistent with the optimal control removing all infected hosts plus hosts that are expected to move into the infected state in that timestep according to the deterministic model. In comparison, the prioritise I controller overspends significantly (the number of removed hosts is much lower than the culling level) and removes the infection more reliably. For the challenging system, the control strategies for MPC and priority based control are very similar when viewed at a system level and both use all of the available resources until the end stages of the outbreak.

## Scaling to larger metapopulations

Fig 6 shows the results for the 4x4 grid. For the controllable system, the priority based methods are still able to eradicate disease quickly and hence they outperform the MPC control. It is interesting to note that the MPC strategies for the controllable system result in a general slowing down of the epidemic without eradication. This is a valid approach within the reward function we have defined. However, there are other costs around an ongoing epidemic which are more difficult to capture in formal reward function and would likely be higher in these scenarios. The intermediate cost scenarios show the priority based control can be advantageous even when swift eradication of the disease is more challenging. The priority based control achieves disease eradication before the epidemic peak in a significant fraction of cases. In comparison, in this set of simulations, the MPC never achieves eradication within the first few critical control steps and the majority of the eradication times are in the second half of the simulation, after the epidemic would have been expected to have peaked. In cases where eradication is straightforward within the available budget, it is reasonable to argue that optimisation is not necessary. However, these intermediate cases show an example where MPC-based optimisation does not get close to eradication but where eradication is still feasible when resource is applied more appropriately for the underlying stochastic system. This is illustrated further with example control and system trajectories in supplementary material (Supporting information). Similar to results on the 2x2 systems, the MPC control has outperformed any of the priority based methods on the 4x4 challenging system. For cases where the priority based methods are unable to achieve local eradications, they can be outperformed by the optimal control but with less overall benefit from any form of control or optimisation.

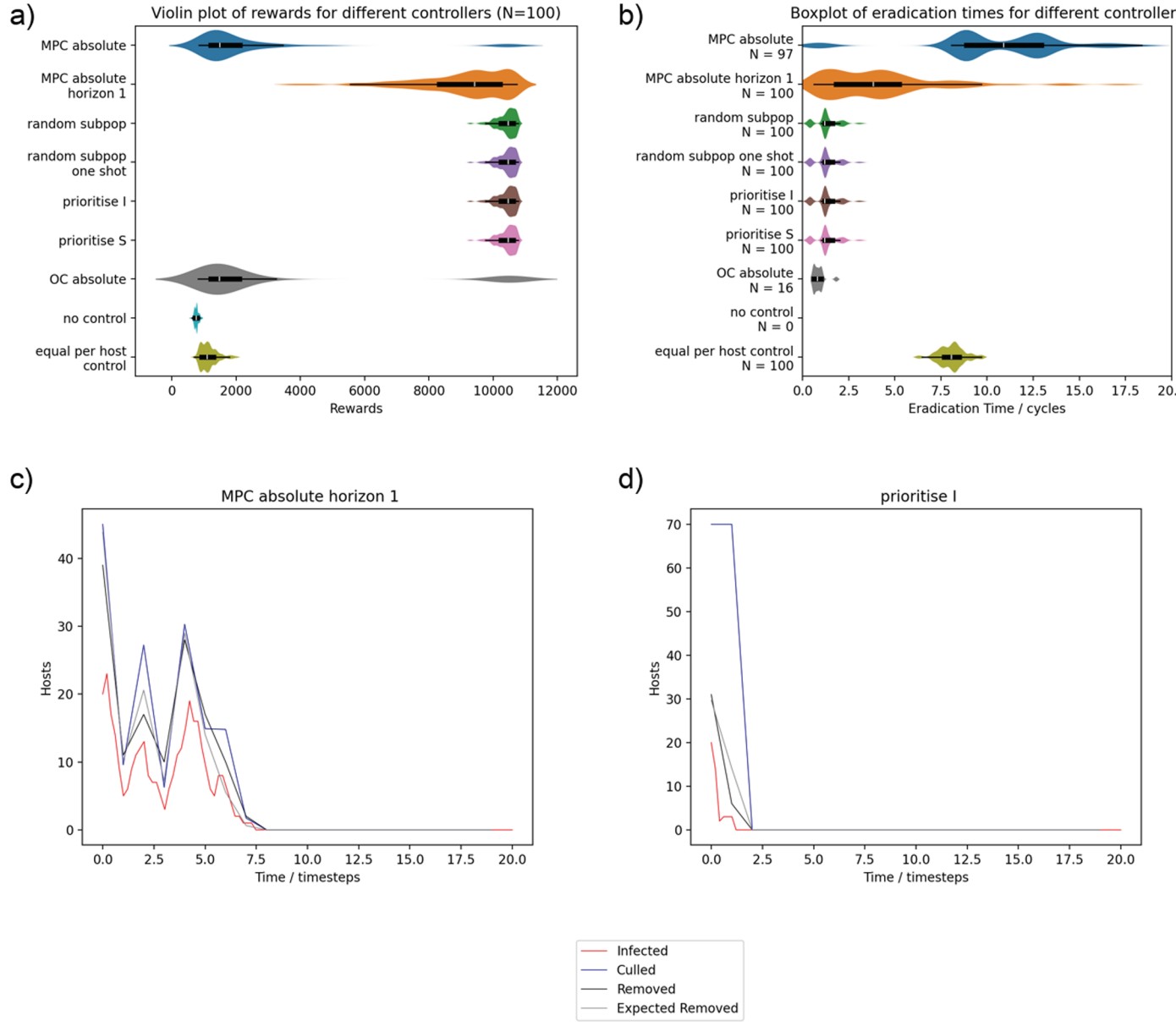

**Fig 4. Controller performance for the controllable 2x2 system.** Violin plot of rewards (a) and eradication times (b). The comparison shows priority list and MPC controllers alongside equal per host control and no control references. The equivalent OC controller is also shown in grey for comparison. The priority list controllers outperform MPC – even when it is using the shortest control horizon – but it does not appear any of the priority list controllers are different to the others. The height of the plots is scaled separately for each of the controllers. Inner box and whisker plot definitions are as per Fig 2. 100 simulations were run for all experiments but the number of samples included in the plot (*N*) is shown separately for each of the controls on the eradication time plot because the disease was not eradicated in all simulations. (c) shows the number of infected hosts and the culling control summed across all subpopulations for an example simulation for MPC absolute horizon 1. The "removed" values shown are not cumulative – they are the number of hosts that died of the disease or were culled within that timestep. The "expected removed" line is an estimation of how many hosts would have been expected to be removed by the control correcting for hosts dying naturally (adding the average of $\gamma I$ across the control step to the specified culling rate) and times during the control period when there were no hosts to cull (for each subpopulation, subtracting the control rate multiplied by the duration when $I$ was zero in that control step). The optimal control removes all infected hosts in the system for each timestep but no more. (d) is an equivalent plot for the prioritise I controller showing the controller overspending to remove the infection more reliably.

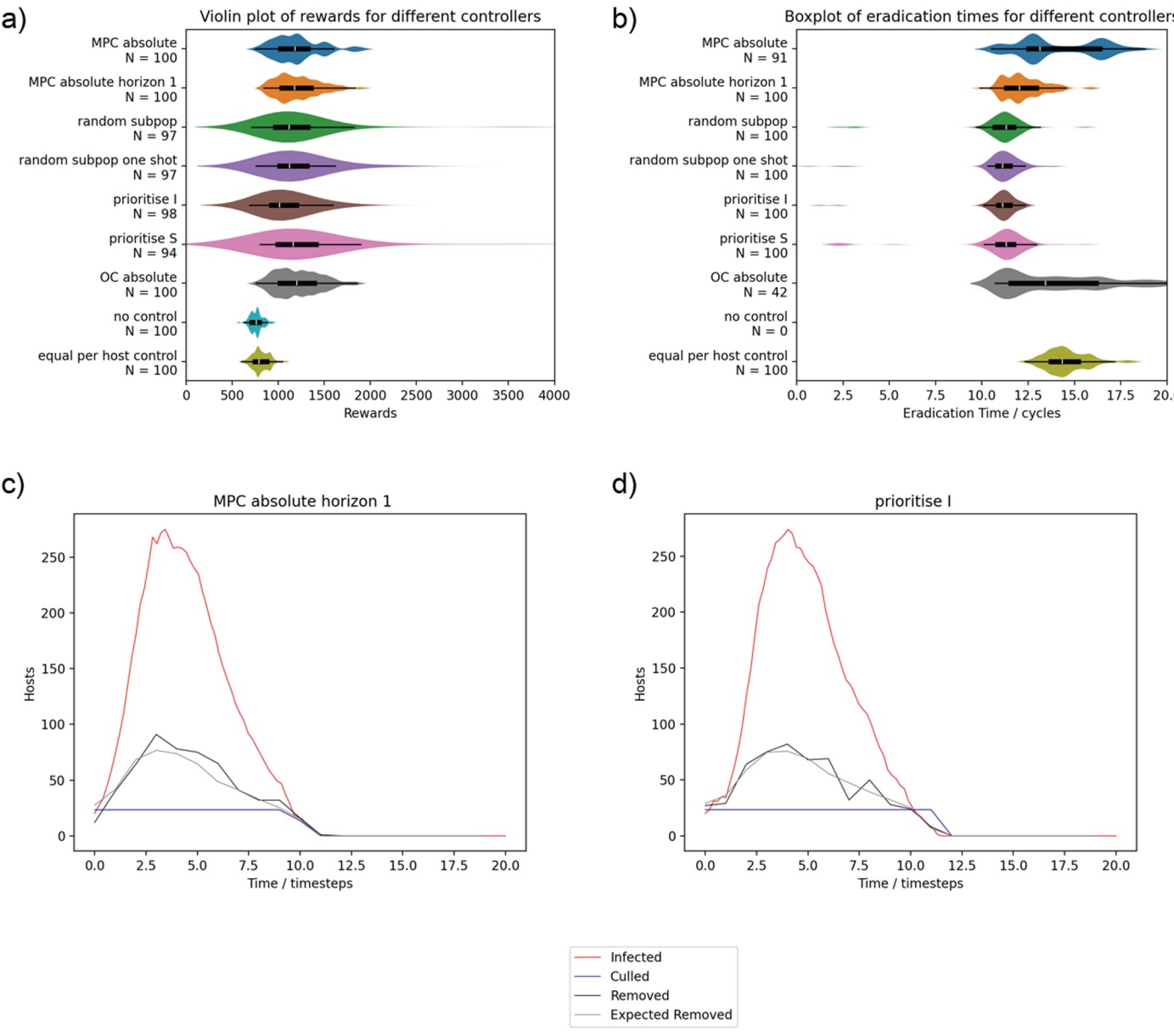

**Fig 5. Results summary for the 2x2 challenging system**. Equivalent to Fig 4. When eradication is not feasible, MPC and optimal control both outperform the priority based controllers (although by a smaller margin). Again, the way in which the subpopulations are prioritised for the priority based controllers appears to be unimportant. The example disease progress curves for the overall system show that both optimal control and priority based controls are using all the available control resource until close to the end of the outbreak. The violin plot of rewards (a) is shown excluding the small number of high scoring cases (reward = 8,000 to 10,000). Each row is a summary of 100 simulations and N is the number of datapoints displayed within the range of the plot.

We also note that the control strategies resulting from both optimal control and from the priority based control are not necessarily realistic in the case of more challenging epidemics as they remove the final infected hosts after the epidemic peak. This results in a smaller epidemic overall and so satisfies the simplistic reward function and model used here but would be an unlikely strategy for real world management as other more complex dynamics around host regeneration and selection would come to dominate the system; it is also possible that control would have been abandoned for an epidemic so manifestly out of control.

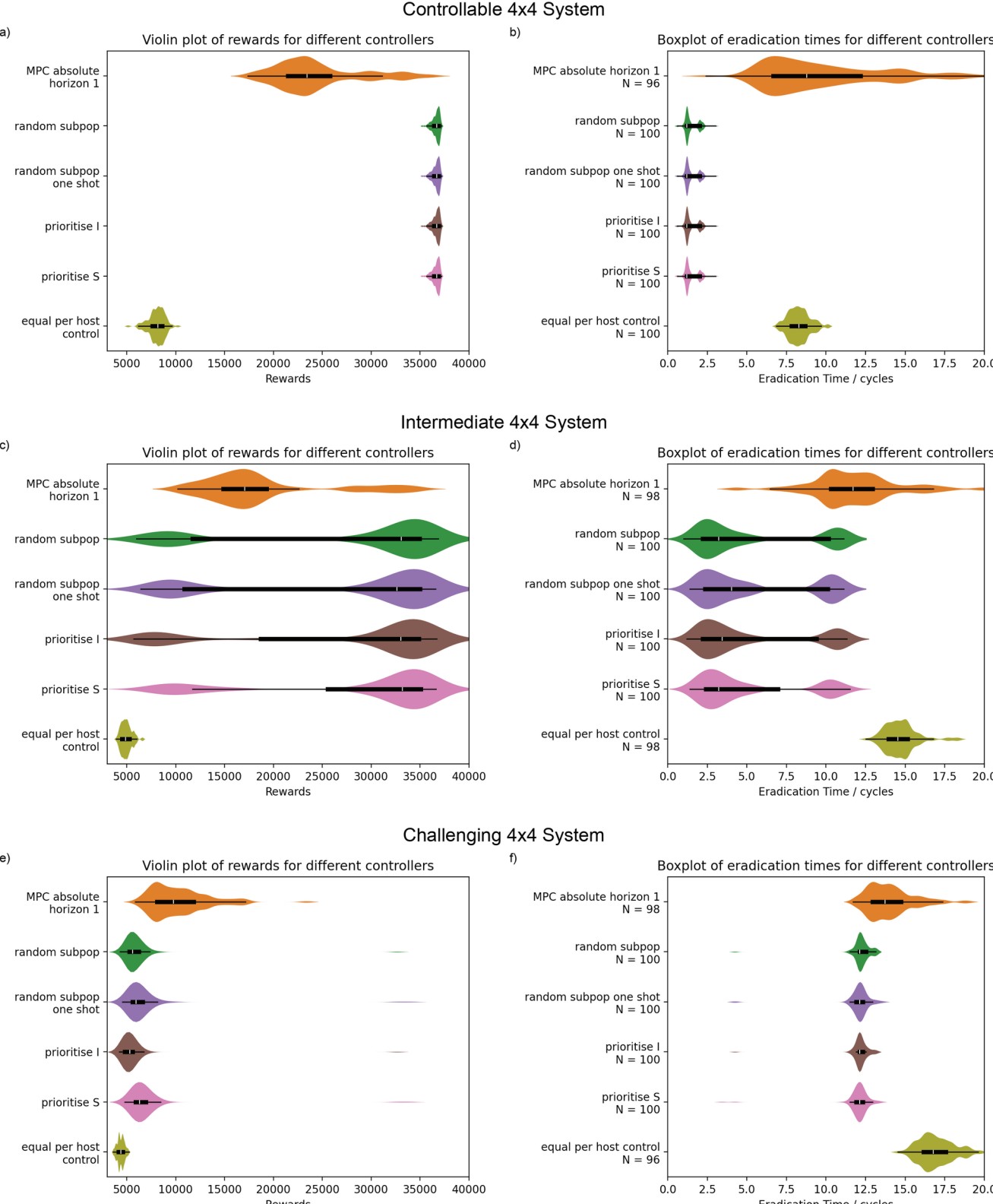

**Fig 6. Performance results for a 4x4 grid.** Sample size of 100 simulations per row, *N* is the number of results within the range of the plot. For the controllable system (a and b), the results are equivalent to the small system with eradications from the priority based methods allowing them to significantly outperform the MPC control. For the intermediate system (c and d), the priority based methods are able to eradicate the disease within the first few control

steps in a significant fraction of the simulations but the optimal control is always removing the final infected hosts after the epidemic peak. For the challenging system (e and f), the MPC control has done better than any of the priority based methods. However, in this case, the overall benefit of any of the optimised methods is relatively small compared to the equal per host baseline.

## Discussion

### Context

To understand why we are considering the optimisation of a stochastic epidemic system using a deterministic model, we first have to understand why optimising the stochastic model is difficult. In simpler systems, it is feasible. For example, [46] directly optimises an MDP analytically balancing the costs of isolation and infection for an SIR model by controlling isolation of infected hosts. However, this approach has not been shown to extend to larger, more complex systems. MDP optimisations under the labels of "stochastic dynamic programming" or "stochastic optimal control" have been investigated in ecological and epidemiological contexts [22]. However, standard approaches for optimising MDPs such as value iteration scale badly with the size of the state and action spaces. For example, the number of states in a metapopulation SIR model with $N$ hosts in each of $M$ subpopulations scales as $O(N^{(3M)})$. This limits applications to cases where status is abstracted to an infection status per subpopulation rather than a detailed host count per compartment [47].

In early work, primarily during the 1970s and 80s, optimal control was explored for simple models and controls where it is possible to determine stochastic optima analytically and compare them with the optimum for the equivalent deterministic system. In general, this was done by determining a policy – a recipe for the action that should be taken dependent on the current state – and comparing points in the state space where the optimal behaviour changes for the stochastic and deterministic versions of the systems. For example, in a simple, abstract SIR model where the only control option is to isolate infective individuals, the trade off between the economic costs of infections and the costs of isolation can be optimised by modelling as stochastic or a deterministic system. The stochastic system was shown to be easier to control in that it was optimal to isolate the population in only a subset of the states where isolation was required in the deterministic system. However, there was no limit to the rate at which the population could be isolated in the model and so the optimal policy is determined by a comparison of the cost to isolate everyone immediately and end the epidemic vs the expected ongoing cost of the disease. The policy is then to isolate everyone if infection falls below a certain level (with the level determined by the remaining number of susceptibles) and we would not expect to see the same issues we have seen in the trajectory based solutions presented here [48]. In work using a similar model but optimising the final number of immunised and recovered hosts, it was simply shown that the form of policies was the same between stochastic and deterministic approaches [49]. [46] investigates a Susceptible-Infected-Removed (SIR) system but assumes that costs are proportional to the rate of control and so the policy of "isolate everyone now" is not available. In this case, using the deterministic policy on a stochastic system for cases with low numbers of infected and high numbers of susceptibles could cause control to be abandoned while eradication was still more efficient. In the dynamic model, cases would be increasing too fast and it would be unable to keep the epidemic under control (hence it is better to let the epidemic run its course). However, in the stochastic case elimination may still be viable. This does not contradict our investigations here because we are highlighting the problem that the deterministic control will not eliminate more infected hosts than it predicts there will be. Even if it is easier to eliminate in the stochastic case due to the possibility of random extinctions, it is still not sufficient to do "just

enough" according to the deterministic model. The approach of comparing policies was not continued to more complex systems likely due to analytical and computational complexity (i.e. the curse of dimensionality [50]).

For these policy based approaches, it could be argued that the stochasticity is accounted for by the continuous time feedback behaviour of the policy. For example, the policy might be "treat subpopulation 3 at the maximum allowed level until no infected hosts remain and then switch to subpopulation 4". The main limitation to this approach is that it is so far not possible to scale the analysis to larger numbers of subpopulations. However, these approaches also assume the manager has fine grain control of the timing of any change of management actions and, in reality, these types of policy may be difficult to implement. In comparison, literature on numerical solutions focuses on generation of solutions in the form of control values as a function of time. In this context, optimal control of epidemics has been extended to MPC in [51] and [7] but neither address the issues we have highlighted around eradication. [51] is aiming to eradicate disease but runs both optimisation and evaluation on a deterministic model. MPC is used solely to account for parameter uncertainty rather than state uncertainty. Hence, it it is questionable if this is a valid way to evaluate effectiveness of eradication because the deterministic models used for evaluation do not accurately model extinction behaviour. The scenario selected for [7] was not aiming for eradication. However, even local eradications were disincentivised due to the cost structure which made the cost of culling proportional to the proportion of infected hosts removed rather than the absolute number.

In addition to stochastic optimal control, there are a wide range of approaches for addressing uncertainty and noise in optimal control problems. However, none of these are targeting solutions to the issue presented here. The two main groups of approaches are robust optimal control and methods similar to Linear Quadratic Gaussian control which take account of system stochasticity using additive noise. Robust optimal control is solving a different problem. It focuses on scenarios where the system needs to remain within certain constraints or ensure adequate worst case behaviour when we have imperfect knowledge of model parameters and/or system state [52]. In comparison, in the problem presented here, we are not aiming to keep the worst case within limits and, in fact, would rather we reached the constraint limit of zero infected hosts as quickly as possible. The constraints in our optimisation are required to keep the model within realistic areas of the state space and do not represent "nogo" areas in the real world. In addition, in order to separate issues related to parameter estimation from more fundamental control behaviour, we have assumed we are in the limiting case in which the model parameters are perfectly known and the system state is perfectly known at intervals. The issues we highlight therefore arise due to the stochastic evolution of the system between inspections. Approaches looking at how far the trajectories could diverge between observations and accounting for this in the optimal control is an interesting area for future investigation. Previous work has addressed these kinds of problems by assuming a form of additive noise (i.e., by converting the epidemic system to an SDE form) [53]. However, epidemic models using SDEs or other forms of additive noise are more difficult to reason about as representations of the real physical system compared to models based on Poisson processes, they do not represent dynamics well when the numbers of infected hosts are small [5] (Sect 6.2) and the best way to formulate the SDE to represent reality is not widely agreed or understood [54]. Hence, further fundamental epidemiological work would be needed before these models could be meaningfully used to optimise control applications.

The eradication situation we have presented is particularly stark. However, the results applying these controls to stochastic models highlight issues that could occur in more complex systems – both missed opportunities to optimise control and cases were control may be less effective than predicted in a significant fraction of cases due to stochastic variation and

a finite time resolution of control. In some work [30,34,37] there is no opportunity to eliminate the disease but, in cases where the infected or vector population is held at low levels for long periods, there is a chance for stochastic effects to be significant and we would expect a wider range of outcomes than indicated by the continuous calculations. In [36], trajectories are presented where the number of hosts is close to or less than 1. In these cases, stochastic effects are likely to be very significant and the control may be overestimated as the possibility of stochastic extinctions has been ignored. In applications where a policy is generated [35], it would be informative to see the impact of more realistic monitoring schedules on the efficacy of the controls when implemented in a stochastic system with different numbers of hosts.

## Potential challenges

There are several lines of argument that could be put forward to challenge this work. It would be reasonable to argue that it is not appropriate to apply a continuous, deterministic optimisation to a stochastic model. However, the underlying assumption is that the stochastic model is a more realistic (although still far from perfect) representation of real life and that if the optimised strategies do not work on the stochastic models, they are unlikely to be optimal in deployment.

It could also be argued that both optimal control and MPC are superior because they are able to give control instructions for a greater length of time after each state measurement compared to the priority based controls. However, the priority based control was presented as a relatively simple way to highlight the issues with the optimal control approach rather than necessarily as the true optimal solution. It is possible to think of other related approaches e.g. where the expected state of the system after control is extrapolated and the immediate optimisation approach repeated for control steps where no observation is available. This does not change the fundamental problem that the optimal control is optimising for a future that is unrealistic. Similarly, if the frequency of surveying was increased relative to the rate of spread of the epidemic then we would expect the discrepancies for the optimal control and MPC to decrease as the determinstic and stochastic models would have less opportunity to diverge. However, the simpler algorithms can be effective with the lower frequency surveys which may be more appealing for real world decision makers. It should also be noted that practical inspection cannot be continuous – there will always be a finite rate of surveying – and so the effects of deterministic model divergence will always be present to some extent.

It is plausible that the "softer" strategies proposed by more constrained optimal control – such as the proportional optimal control – may practically be more robust to imperfections in inspection and control in some situations. However, this would have to be demonstrated on a case by case basis because there is no mechanism in the controller mathematics to specify more conservative control. It may have found a robust solution but it has not been set up to optimise for that robustness. Similarly, it may be possible and pragmatic to bias the optimal control towards scenarios where local or global eradication is likely by using a different reward formulation. However, we judged this to be unlikely to work well because the optimal control is fundamentally bounded on the levels of control it can apply. The optimisation can have arbitrarily large incentives to prefer states with zero infected hosts but it will still be constrained by the deterministic model and so will be unable to apply sufficient control to achieve them in *all* cases. Additionally, this would require tuning on a case by case basis and it becomes difficult to describe and understand what the control is actually optimal for.

The effects described were only demonstrated on small, grid-shaped systems. In realistic, plant focused systems, the subpopulations would often be formed from a raster grid over a landscape [3,19,55–58] and so this can be viewed as a minimal version suitable for easy

experimentation with MPC. In real use cases, iterations on the control strategy might occur every few weeks at most and so the runtimes of MPC iterations on larger systems could be acceptable. However, this is challenging for the purposes of evaluation. The largest 4x4 system was chosen to be small enough to be computationally tractable without significant manual tuning of the optimisation code but large enough to demonstrate any interaction or second order effects that might occur in more realistic systems. The more general concept of minimising fatalities for a metapopulation using limited resources to move hosts from infected or susceptible compartments to removed (or recovered for human and animal systems) is applicable across a wide range of different pathosystems and scenarios. For example, optimisation of vaccination allocation across different locations or age groups or allocation of limited culling resource in the management of a transmissible animal disease [21,59,60].

A different and equally reasonable modelling approach would be to apply all of the control for each subpopulation at the start of each control step. This more accurately captures real world control activities – i.e. it is more likely that control in a given subpopulation would happen in a short period of time while a control agency was in that area. However, this is further away from the model used internally to the deterministic optimal control and so is open to criticism that any degradation in performance is due to this difference in modelling.

Focusing on eradication as a strategy has its own challenges. These have been summarised for human disease in [61] but are more commonly approached on a per-pathosystem basis [62–64]. The overarching theme is that the end stages of eradication are more difficult than managers expect. This is due to the challenges of finding the final pockets of infection in a system with large numbers of susceptible hosts where the disease can potentially spread very quickly. In the model studied here, the economics are arguably oversimplified to a flat cost per host management action. However, the most obvious alternative, using costs dependent on the proportion of infected hosts removed as per [7], effectively precludes eradication and so some, more complex intermediate approach would be required. The scenario studied here also assumes perfect information in order to allow integration with optimal control. More pragmatic recommendations indicate that delay in detection is the key limitation preventing eradications being successful in the context of plants [65] (Sect 23.5). This directs future work towards optimisations with incomplete information [66] and with more realistic modelling of eradication dynamics and costs [67].

## Conclusion

Deterministic, ODE based, continuous optimal control has been widely applied to the problem of epidemic control. However, the deterministic models are known to be poor approximations to reality when the number of infected hosts in a population or subpopulation is low. Organisations which make management decisions about disease control will often prefer disease eradication in cases when it may be feasible and so scenarios where the numbers of infected hosts are low should not be ignored. This work has illustrated that deterministic, ODE based, continuous optimal control can be outperformed by much simpler, priority based controllers for a number of representative, small, stochastic epidemic systems and, hence, we conclude that they are not always a useful approximation to the true optimal control. This is because the controller expects the system to evolve in a particular way and in cases where eradication is possible, the model and the stochastic system (either model or reality) can diverge significantly. The models used by the optimisers cannot account for the different levels of controls needed to achieve eradication in different instances of the stochastic system. This is not solved by using MPC. This was demonstrated for a particular simple

metapopulation model based on the control options available in a plant disease control scenario. However, we would expect similar issues to occur in other cases where local or global eradication are feasible and useful within the system. For practitioners looking to control an epidemic, a plausible and logistically straightforward approach is to focus on removing all infected hosts in a given area before moving to the next. This would lead to similar control actions to the priority based heuristics. Hence, when the available resources are sufficient to eradicate the pathogen, our work indicates that the simple approach outperforms recommendations based on optimal control theory. Our examples also form a very important cautionary tale – models are often used in practice to demonstrate that eradication is no longer feasible and to justify moving to an approach of limiting spread. Optimal control would appear to be a good way to evaluate this. However, this work shows that eradication may still be feasible when deterministic optimal control does not achieve it.

It should be clearly stated that the effects described in this work do not preclude the use of deterministic models for optimal control of epidemics as they can be a good representation of the system in some cases. Deterministic optimal control could be validly used in cases where neither global or local eradication of disease is feasible within the constraints. For example, in [68], there is a constant infectious pressure on the system. This situation will be more common in cases with low levels of resource although it may still be optimal to consider local eradication in more isolated subpopulations or groups. Hence, in many cases, it may be difficult to determine or calculate if a scenario has relevant, reachable eradication states.

The allocation of all the control resource was an important feature of the priority based control. In a stochastic system where the control is represented as a rate and where the amount spent on control in each control step is represented as a bound rather than as part of a reward function, any optimal policy for cases where there are infected hosts in the system will always spend all of the budget. We can argue this by thinking about an arbitrary candidate for an optimal policy that does not spend all of the budget. Within the assumptions, spending more of the budget on culling compared to this arbitrary policy cannot make the performance worse (as culling does not affect the reward or increase the rate of hosts transitioning from susceptible to infected). However, if extra culling resource is added to nodes containing infection, the infection will be cleared more quickly and hence there is less probability that it will spawn more infections. Hence, the candidate optimal policy will always be improved upon by a policy which uses all the remaining budget.

More generally, this work highlights a tension between optimisation and stochasticity – the objective to use minimal resources to achieve a target outcome and the need to manage risk in a system where a range of outcomes are possible for any given control. For example, in cases where the objective is to manage hospitalisations below a critical threshold [69], a determinstic model may optimise the infected population to sit consistently on the threshold of system overload. In a stochastic version of the same system, we would expect a range of outcomes and so a significant fraction of the results would be expected to break the limit. It is unclear how a deterministic goal could be altered in a principled way to accurately represents the spread of risk. We could chose to add buffers around the goal to ensure it is met more frequently. However, this becomes a parameter that must be determined for each scenario to achieve the desired trade off between risk and optimality and does not allow for differing levels of buffering as the situation progresses. Hence, the work presented here could also be viewed as evidence that absolute optimisation goals (as opposed to probabilistic ones) may not be appropriate in many epidemic scenarios [21].

## Supporting information

**S1 Appendix. Individual system and control trajectories for MPC and priority based control.** Plots and descriptions of the trajectories of infected hosts and applied control for a subset of the subpopulations in the 4x4 intermediate system.
(PDF)

## Author contributions

**Conceptualization:** Rachel Russell, Nik J. Cunniffe.

**Data curation:** Rachel Russell.

**Formal analysis:** Rachel Russell.

**Funding acquisition:** Nik J. Cunniffe.

**Investigation:** Rachel Russell.

**Methodology:** Rachel Russell.

**Project administration:** Rachel Russell.

**Resources:** Nik J. Cunniffe.

**Software:** Rachel Russell.

**Supervision:** Nik J. Cunniffe.

**Validation:** Rachel Russell.

**Visualization:** Rachel Russell.

**Writing – original draft:** Rachel Russell.

**Writing – review & editing:** Rachel Russell, Nik J. Cunniffe.

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
