## [Decision Letter · Decision Letter 0]

15 Aug 2024

Dear Mrs Russell,

Thank you very much for submitting your manuscript "Optimal control prevents itself from eradicating stochastic disease epidemics" for consideration at PLOS Computational Biology.

As with all papers reviewed by the journal, your manuscript was reviewed by members of the editorial board and by several independent reviewers. In light of the reviews (below this email), we would like to invite the resubmission of a significantly-revised version that takes into account the reviewers' comments.

We cannot make any decision about publication until we have seen the revised manuscript and your response to the reviewers' comments. Your revised manuscript is also likely to be sent to reviewers for further evaluation.

Sincerely,

Varun Dutt, Ph.D

Academic Editor

PLOS Computational Biology

Thomas Leitner

Section Editor

PLOS Computational Biology

Reviewer's Responses to Questions

**Comments to the Authors:**

Reviewer #1: In this manuscript, the authors highlight the use of stochastic models over deterministic continuous optimal control models for epidemic control problems. The Authors consider cases of small populations of infected hosts where eradication is a feasible scenario, to show the application of stochastic models and comparisons to simpler priority-based controls.

I suggest the following edits for revision

1. I suggest rewriting the author summary to make it more clear. In Author summary:

- Optimisations –spelling check

- Allocating limited resources to reduce the impact of an epidemic is important across human, animal and plant health. – not clear

- For example, deciding which infected plants to remove first to reduce ecosystem or agricultural losses from an invasive plant disease [ is important/critical?] – incomplete sentence.

- A common approach to these optimisations is to model the epidemic as a totally

predictable (deterministic) process and modern algorithms allow these optimisations to

scale to large, real-world problems (e.g. allocating resources across a landscape).

Which optimizations are being referred here?

2. Introduction - Management aims vary dependent on the scenario and pathosystem in question (see [6] for a review of the equivalent problem in the context of established invasive species). Which management?

3. There are Hyphens (-) inside a few of the sentences throughout.

4. It is mentioned that the plausible assumptions is that human decision making happens at discrete intervals and that there may be some delay between the decision making and the action being carried out. In that case, is this delay incorporated in the ODE’s or anywhere else? If not, give reasons

5. FIGURES placements need to be fixed. Some of them overflowing from the page and not labeled

6. Consistency in references to citations

Reviewer #2: Review uploaded as attachment.

**Have the authors made all data and (if applicable) computational code underlying the findings in their manuscript fully available?**

Reviewer #1: Yes

Reviewer #2: Yes

PLOS authors have the option to publish the peer review history of their article (what does this mean?). If published, this will include your full peer review and any attached files.

Reviewer #1: No

Reviewer #2: No
---

## [Decision Letter · Decision Letter 1]

10 Jan 2025

Dear Mrs Russell,

We are pleased to inform you that your manuscript 'Optimal control prevents itself from eradicating stochastic disease epidemics' has been provisionally accepted for publication in PLOS Computational Biology.

Best regards,

Varun Dutt, Ph.D

Academic Editor

PLOS Computational Biology

Thomas Leitner

Section Editor

PLOS Computational Biology

Reviewer's Responses to Questions

**Comments to the Authors:**

Reviewer #1: All the comments have been addressed.

Reviewer #2: I like the changes you have made since the first review. Specifically, I think you now do a better job situating your work relative to the standard assumptions (deterministic ODEs) and other control literature.

**Have the authors made all data and (if applicable) computational code underlying the findings in their manuscript fully available?**

Reviewer #1: None

Reviewer #2: Yes

PLOS authors have the option to publish the peer review history of their article (what does this mean?). If published, this will include your full peer review and any attached files.

Reviewer #1: No

Reviewer #2: No

---

## [Editor Report · Acceptance letter]

PCOMPBIOL-D-24-00724R1

Optimal control prevents itself from eradicating stochastic disease epidemics

Dear Dr Russell,

I am pleased to inform you that your manuscript has been formally accepted for publication in PLOS Computational Biology. Your manuscript is now with our production department and you will be notified of the publication date in due course.

With kind regards,

Zsofia Freund
